# The P2X7R-NLRP3 and AIM2 Inflammasome Platforms Mark the Complexity/Severity of Viral or Metabolic Liver Damage

**DOI:** 10.3390/ijms23137447

**Published:** 2022-07-04

**Authors:** Chiara Rossi, Antonio Salvati, Mariarosaria Distaso, Daniela Campani, Francesco Raggi, Edoardo Biancalana, Domenico Tricò, Maurizia Rossana Brunetto, Anna Solini

**Affiliations:** 1Department of Surgical, Medical, Molecular and Critical Area Pathology, University of Pisa, Via Roma 67, I-56126 Pisa, Italy; chiara.rossi@unipi.it (C.R.); mariarosariadistaso93@msn.com (M.D.); francesco.raggi@unipi.it (F.R.); 2Azienda Ospedaliero-Universitaria Pisana, I-56126 Pisa, Italy; antonio.salvati@ao-pisa.toscana.it; 3Department of Translational Research and New Technologies in Medicine and Surgery, University of Pisa, I-56126 Pisa, Italy; daniela.campani@unipi.it; 4Department of Clinical and Experimental Medicine, University of Pisa, Via Roma 67, I-56126 Pisa, Italy; e.biancalana@yahoo.it (E.B.); domenico.trico@unipi.it (D.T.)

**Keywords:** P2X7 receptor, NLRP3, AIM2, non-alcoholic fatty liver disease, HCV infection

## Abstract

P2X7R-NLRP3 and AIM2 inflammasomes activate caspase-1 and the release of cytokines involved in viral-related liver disease. Little is known about their role in non-alcoholic fatty liver disease (NAFLD) and non-alcoholic steato-hepatitis (NASH). We characterized the role of inflammasomes in NAFLD, NASH, and HCV. Gene expression and subcellular localization of P2X7R/P2X4R-NLRP3 and AIM2 inflammasome components were examined in histopathological preparations of 46 patients with biopsy-proven viral and metabolic liver disease using real-time PCR and immunofluorescence. P2X7R, P2X4R, and Caspase-1 are two- to five-fold more expressed in patients with NAFLD/NASH associated with chronic HCV infection than those with metabolic damage only (*p* ≤ 0.01 for all comparisons). The AIM2 inflammasome is 4.4 times more expressed in patients with chronic HCV infection, regardless of coexistent metabolic abnormalities (*p* = 0.0006). IL-2, a cytokine playing a pivotal role during chronic HCV infection, showed a similar expression in HCV and NASH patients (*p* = 0.77) but was virtually absent in NAFLD. The P2X7R-NLRP3 complex prevailed in infiltrating macrophages, while AIM2 was localized in Kupffer cells. Caspase-1 expression correlated with elastography-based liver fibrosis (r = 0.35, *p =* 0.02), whereas P2X7R, P2X4R, NRLP3, Caspase-1, and IL-2 expression correlated with circulating markers of disease severity. P2X7R and P2X4R play a major role in liver inflammation accompanying chronic HCV infection, especially when combined with metabolic damage, while AIM2 is specifically expressed in chronic viral hepatitis. We describe for the first time the hepatic expression of IL-2 in NASH, so far considered a peculiarity of HCV-related liver damage.

## 1. Introduction

The inflammasome, a multiprotein complex located in the cellular cytosol, assembles in the presence of damage or danger stimuli (PAMPs or DAMPs) to safeguard the integrity of the cells [1]. In the last years, different types of inflammasomes have been identified and classified according to their structural characteristics or ligand specificity [2]; one of the most studied is NLRP3, preferentially but not exclusively activated through the P2X7 receptor (P2X7R), a transmembrane receptor sensitive to extracellular ATP. P2X7R plays a role in maintaining intracellular calcium homeostasis and is involved in various cellular regulatory functions like the release of inflammatory cytokines and chemokines, survival and differentiation of T lymphocytes, activation of transcription factors, and cell death through apoptotic and necrotic pathways involving Caspase-1 [3]. The P2X receptorial system includes the P2X4 receptor (P2X4R) that displays a marked homology with P2X7R, being another preferential receptor involved in the activation of NRLP3 inflammasome, although sensitive to different concentrations of extracellular ATP [4]. Absent in melanoma 2 (AIM2), inflammasome is another key inflammatory platform, preferentially activated by double-stranded DNA or RNA [5,6]. The next step in the AIM2 activation sequence is the recruitment of the ASC, the common adaptor of most inflammasomes [7]; the final common pathway is, again, the Caspase-1 activation, followed by proinflammatory cytokines maturation.

The liver targets both viral and metabolic insults; these conditions are characterized by a massive inflammatory process, sometimes leading to irreversible organ damage [8]. Several studies conducted in mouse and cellular models have documented a significant mRNA expression of the NRLP3-inflammasome components and P2X7R in case of HCV infection [9,10] and non-alcoholic steatohepatitis (NASH) [11,12], although studies addressing the link of P2X7R and P2X4R with non-alcoholic fatty liver disease (NAFLD) are scanty. Furthermore, AIM2 is expressed in lymphomonocytes and liver tissue of patients with chronic hepatitis B [13,14], which correlates with liver inflammation score but not with fibrosis [15]. The inflammasome activation and subsequent IL-1β release seem to relevantly contribute to the progression of NAFLD to NASH [16]; however, human studies are lacking.

The comparative evaluation of P2X7R and AIM2 expression in viral and metabolic liver diseases and their relationships with the two main hepatic inflammasomes have never been characterized; similarly, their putative predictive role in the clinical progression of the disease and patient’s prognosis is unknown. This work is based on a retrospective study comparing histopathological preparations with patients’ clinical characteristics. Biopsies from patients affected by different hepatic diseases were used to characterize the expression and localization of P2X7R and other components of the inflammasomes in the liver and identify the involved intracellular pathways.

## 2. Results

Table 1 shows the bio-anthropometric phenotype of patients recorded on the day of the liver biopsy.

The clinical characteristics of the two main groups (NAFLD/NASH or HCV) were similar, except for insulin levels that were significantly higher in NAFLD/NASH patients, as expected. Appendix A show the individual liver tissue grading and staging of the patients.

To clarify the possible involvement of the P2X receptorial system in these different clinical conditions, we measured mRNA expression of P2X7R and P2X4R in hepatic biopsies: both receptors were significantly more abundant in subjects presenting HCV infection (Figure 1a,b) compared with NAFLD/NASH. When the patients were divided into subgroups according to their specific disease, we observed that the greatest P2X7R and P2X4R expression was in patients showing the combination of HCV infection and metabolic damage (Figure 1c,d).

NAFLD is characterized by an excessive fat accumulation in the liver; the progression of this condition involves, among other pathways, the NRLP3-inflammasome complex [12,17]; we, therefore, measured NRLP3 and Caspase-1 mRNA expression. Data are shown in Figure 2. Caspase-1 resulted more abundantly represented in hepatic biopsies of HCV patients (Figure 2b); a similar trend was also observed for NLRP3 (Figure 2a). When we stratified the participants into subgroups, Caspase-1 exhibited comparable expression levels in NAFLD, NASH, and HCV-nMet, while it was significantly more abundant in subjects with HCV-infection associated with the metabolic disorder (HCV-Met) (Figure 2d).

The next step was to measure the expression of the absence of melanoma 2 (AIM2) inflammasome in liver biopsies. As expected, it was mainly expressed in patients with chronic HCV infection but was also detectable in subjects with NAFLD or NASH. Data are shown in Figure 3a,c. IL-2, a cytokine playing a pivotal role during chronic HCV infection, was virtually absent in NAFLD and highly expressed in HCV (Figure 3b,d); surprisingly, patients with steatohepatitis (NASH) showed an expression level comparable to that of HCV patients (Figure 3d).

The effect of liver disease on gene expression was confirmed in multivariable models after adjustment for potential confounders (age, sex, BMI): all the inflammasome components (P2X7R, P2X4R, NLRP3, CASP1, AIM2, IL-2) were significantly related to the liver disease (0.0007 < *p* <0.037).

Liver expression of other inflammatory markers like IL-6 and TNFα, often increased during HCV infection [18,19], were significantly higher in HCV subjects; TGFβ did not differ. Data are shown in Appendix A.

Immunofluorescence experiments were performed to define the subcellular localization of these proteins better. As shown in Figure 4a, in sections obtained from needle biopsies, the immunoreactivity for P2X7R was mostly surrounding the vessels (centrilobular vein), suggesting that this receptor could be expressed in cells coming from hepatic circulation, e.g., infiltrating macrophages, although red fluorescence was also observed in the non-centrilobular area (Figure 4a and Figure 5), making this hypothesis worthy of confirmation. The aspect of P2X7R at immunofluorescence was slightly different in HCV+ and HCV- patients, being revealed mainly around vessels as fibril aggregates in the former and more widespread in the latter; this observation is reported in Figure 5. In experiments conducted with anti-AIM2 antibody, the immunofluorescent signal appeared to be localized near sinusoids (Figure 4b); double staining AIM2/CD68 (the latter selectively marking Kupffer cells) confirmed that AIM2 inflammasome is expressed in the Kupffer cells (Figure 4c).

High magnification images using all combinations of primary antibodies clarified the different cellular localization of the inflammasomes: AIM2, not NLRP3, resulted expressed in CD68^+^ cells (AIM2^+^/CD68^+^ cells: 18.3% and NLRP3^+^/CD68^+^: 0.6%) (Figure 6a,b), while P2X7R-positive cells co-expressed NLRP3 (NLRP3^+^/P2X7R^+^ cells: 27.6%), but not AIM2 (AIM2^+^/P2X7R^+^ cells: 1.2%) (Figure 6c,d).

Thereafter, we explored the relationships between the expression of genes of interest and markers of disease severity. According to more aggressive and severe diseases, liver stiffness resulted significantly higher in HCV-Met than in NAFLD (Figure 7).

In the whole study population, linear correlations emerged between liver stiffness index and hepatic Caspase-1 expression (Figure 8). ALT levels were proportional to P2X7R, P2X4R, NRLP3, Caspase-1, and IL-2 expression (r > 0.20 and *p* ≤ 0.05 for all), while AST levels were proportional to P2X4R, Caspase-1, and IL-2 expression (r > 0.30 and *p* ≤ 0.04 for all) (Figure 8).

Although liver stiffness index was not related to NLRP3 in the whole cohort, a statistically significant (*p* < 0.05) correlation emerged in the HCV-Met group (Appendix A).

Finally, to test whether such different liver expressions of IL-2 would translate into different circulating levels, we measured IL-2 in plasma samples collected on the day of the liver biopsy. IL-2 was 0.83 ± 0.53 pg/mL in NAFLD; 0.88 ± 0.46 in NASH; 1.32 ± 0.79 in HCV-nMet and 1.38 ± 1.12 pg/mL in HCV-Met, without significant group differences. Hepatic and circulating IL-2 did not correlate in the whole cohort (r = 0.12, *p* = 0.44) and subgroups of patients.

## 3. Discussion

This study provides a detailed description of the P2X7R-NLRP3 and AIM2 inflammasomes in the liver of patients carrying different hepatic diseases (viral or metabolic damage). We show here that: (i) the P2X7R-NLRP3 inflammatory platform is more expressed in patients with chronic HCV infection associated with metabolic damage and that P2X4R shows the same behavior; (ii) the AIM2 inflammasome is more expressed in patients with chronic HCV infection, regardless of the coexistence of metabolic abnormalities; (iii) these two types of inflammasome are differentially immunolocalized within the liver tissue; (iv) gene expression correlates with liver fibrosis and with circulating markers of disease severity; (v) IL-2 liver expression, virtually absent in NAFLD patients, seems to discriminate NASH over simple steatosis clearly.

Previous data documented a significant increase in the number of P2RX7^+^ cells in the NASH-affected liver biopsies compared to healthy controls, although the P2RX7 mRNA was similar; this was coupled with increased NLRP3, IL-1β, Caspase-1, and AIM2 mRNA [20]; therefore, our observations are somehow confirmatory of the presence of the P2X7R in resident hepatic cells of patients with metabolic liver disease. On the contrary, we are reporting here, for the first time to our knowledge, a relevant liver expression of this platform components in patients affected by chronic HCV hepatitis and that it might be amplified when NAFLD or NASH coexist with a viral infection. Such human observations have been performed only in peripheral blood mononuclear cells of HCV+ patients [10,21] or in liver tissue samples of HCV-derived hepatocarcinoma [22]; the increased presence of the P2X7R in the liver of HCV+ individuals warrants further exploration of its potential role as a predictive marker.

The relevant expression of P2X7R/P2X4R, NRLP3-inflammasome complex, and Caspase-1 in HCV liver could be fostered by the close interplay between HCV and host lipid metabolism, where numerous factors involved in lipid metabolism are required for HCV replication, virus assembly, and production. Accordingly, chronic HCV infection is associated with dysregulated lipid homeostasis, favoring triglycerides accumulation in the liver. P2X4R is an ionotropic receptor sharing evolutionary origin, several molecular characteristics, and likely functional activities with P2X7R [23,24,25], whose increased expression has been described already in circulating mononuclear cells of HCV+ patients [21] and in the liver samples of adenocarcinoma and ampullary carcinoma [26]. In our study, P2X4R expression followed the same trend as P2X7R, confirming their influence as proinflammatory receptors in cancer progression and prognosis [27]. P2X4R is barely represented in normal liver tissue (24); its relevant expression in liver biopsies of HCV individuals opens a new perspective on its potential effect as a pro-viral agent, reinforcing its role as an important component of the purinergic signaling complex in HCV-induced liver disease [28].

Different types of inflammasome are activated in the liver in response to virus/microorganism infection or toxic substances exposition [29]. AIM2 inflammasome is directly activated by dsDNA from viruses, bacteria, or host cells and can also be activated by RNA viruses [30]. Our data corroborate the relevant role played by AIM2 in virus-related liver diseases [31]. At odds with that observed in animal models, where AIM2 inflammasome activation might contribute to inflammation and progression of NAFLD to NASH [32,33], we could not confirm this observation in humans: in NAFLD and NASH subjects, liver expression of AIM2 was negligible.

A relevant novelty of this report is to describe, for the first time, a specific localization of the inflammasome platforms (P2X7R/NLRP3 and AIM2) in the liver, being the P2X7R/NLRP3 complex mainly surrounding the centrilobular vein and AIM2 localized near sinusoids. Both localizations pertain to inflammatory cells but with a different function: infiltrating macrophages coming from the systemic circulation for P2X7R/NLRP3, residing hepatic specific macrophages (i.e., Kupffer cells) for AIM2. Another interesting point is that such localization is confirmed in all the biopsies we have analyzed, irrespective of the disease (metabolic, viral, or both). Unfortunately, we do not have specimens of the healthy liver to confirm this report in the absence of any liver disease, but we are prone to hypothesize that the P2X7R/NLRP3 complex might be virtually absent in the healthy liver, being activated only in the presence of strong inflammatory noxae, as previously reported in other organs and tissues [34,35].

IL-2 is predominantly secreted during chronic HCV infection, where it promotes tissue damage and fibrosis [36]. Unexpectedly, we found that IL-2 expression in liver tissues was increased in NASH, but not in NAFLD, toward the same levels observed in HCV patients. This finding qualifies IL-2 as a non-HCV-specific marker of liver necro-inflammation and corroborates a pathogenetic role of the activation of the IL-2 pathway in the progression of liver fibrosis, regardless of the nature of the primary insult (viral vs. metabolic), as previously supported by indirect evidence [37,38]. Hence, we measured serum IL-2 to assess whether it may discriminate simple steatosis from NASH, but no differences emerged between groups. This result may be explained by the small sample size, which is unlikely given that negative findings were also observed in a larger (*n* = 648) cohort of NAFLD patients [36], or by our observation that circulating IL-2 levels do not reflect local IL-2 expression, and presumably concentration, within the liver. Multivariate analysis confirms the relationship between inflammasome components and liver disease; although based on the current knowledge, it is likely that the increased expression of genes involved in inflammatory processes is causally linked to the pathogenesis of NASH or chronic HCV hepatitis, this cannot be firmly established from correlation analyses on cross-sectional data.

The strength of the present study resides in the definite diagnosis provided by the biopsy and the chance to correlate liver stiffness index and circulating markers of disease severity with a liver expression of these biomarkers. Limitations include the relatively small cohort size, the lack of liver biopsies during the patients’ follow-up, and the absence of liver specimens from healthy individuals, not allowing to conclude both quantitative expression and topography of the two inflammasomes in a normal liver. Moreover, we should point out that liver biopsies in these subjects were exclusively justified by clinical indications; the available material was scarce and did not allow for more sophisticated technical approaches.

In conclusion, both P2X7R and P2X4R seem to play a role in liver inflammation accompanying chronic HCV infection, especially when combined with metabolic damage, while AIM2 appears specifically expressed in chronic viral hepatitis. The P2X7R/NLRP3 inflammasome surrounds the centrilobular vein, likely marking an inflammatory infiltrate, while AIM2 is localized in Kupffer cells near sinusoids. We also describe the hepatic expression of IL-2 in NASH for the first time, so far considered a peculiarity of HCV-related liver damage. These observations should be confirmed in larger cohorts of patients with available liver tissue samples, and prospective studies are needed to provide evidence on the prognostic, predictive value of such inflammatory markers.

## 4. Materials and Methods

### 4.1. Participants

In this retrospective, observational study, we reconsidered the liver biopsies of 46 consecutive treatment-naive patients with chronic hepatitis C and non-alcoholic fatty liver disease (NAFLD) with or without steatohepatitis (NASH) who underwent a liver biopsy at the Hepatology Unit of the University Hospital in Pisa in the years 2005–2017 because of chronic liver disease associated with fatty liver and chronic hepatitis C (CHC) virus infection, without clinical and ultrasound signs of cirrhosis. Indication to biopsy was set on clinical criteria to pursue a correct diagnosis and staging of the disease. We included in the study only asymptomatic patients with chronic hepatitis C and NAFLD without other known causes of liver disease. They were stratified into four groups: (A) NAFLD without steatohepatitis (NAFLD); (B) NAFLD with steatohepatitis (NASH); (C) Chronic Hepatitis C without metabolic alterations (HCV-nMet), and (D) Chronic Hepatitis C with metabolic alterations (HCV-Met).

For each patient, anamnestic data, pharmacologic treatment, and biochemistry were collected at the time of the liver biopsy. Complete blood count, serum liver enzymes (aspartate aminotransferase [AST], alanine aminotransferase [ALT], gamma-glutamyl transferase [GGT]), liver function tests (albumin, total and direct bilirubin, prothrombin time), lipid profile, glucose were measured according to standard laboratory procedures at the central laboratory of our hospital. Fasting insulin concentration was measured by chemiluminescent immunoassay (LIAISON^®^, DiaSorin S.p.A., Saluggia, Italy). Serum and plasma aliquots stored at −20 °C at the time of biopsy were used for further determinations.

The study protocol was approved by the Ethics Committee of the University of Pisa (Comitato Etico Area Vasta Nord Ovest (CEAVNO) 1179/2016); all patients gave their informed consent for using residual histologic material from liver biopsy for scientific purposes.

### 4.2. Liver Stiffness Measurement (LSM)

Before undergoing a biopsy, all patients received liver ultrasonography, performed after overnight fasting by FibroScan (Echosens, Paris, France) and an M probe equipped with the controlled attenuation parameter (CAP) technology for assessing hepatic steatosis by a single trained physician. Each patient’s LSM was considered adequate if it included at least 10 valid measurements, with a success rate >60% and measurement variability <30% of the median [39]. In this cohort, LSM strongly correlated with biopsy-derived stage scores in both NAFLD (r = 0.60, *p* = 0.003) and HCV patients (r = 0.76, *p* < 0.0001); therefore, FibroScan values were used to examine the influence of each gene expression on liver fibrosis across different scoring systems.

### 4.3. Liver Biopsy

Sections of formalin-fixed and paraffin-embedded liver specimens were stained with hematoxylin and eosin. The histological examination was performed by an expert liver pathologist who graded inflammation and fibrosis according to Ishak score [40] in CHC liver biopsy and Brunt score for NAFLD [41].

### 4.4. Gene Expression

Total RNA was extracted from formalin-fixed paraffin-embedded (FFPE) liver sections using the RecoverAll Total Nucleic Acid Isolation kit (Ambion, Thermo Fisher Scientific, Waltham, MA, USA). Briefly, 5–10 sections (10 µm thick) were cut from FFPE samples; they were deparaffinized with xylene, rehydrated in decreasing alcohol series, and processed according to manufacturing advice. RNA (1 µg), retrotranscribed with SuperScript Vilo kit (#11754 Thermo Fisher, Waltham, MA, USA), was analyzed using the Eco Real-Time system (Illumina Inc., San Diego, CA, USA). Transcripts were evaluated by the following TaqMan Gene Expression Assay (Thermo Fisher): P2X7R, Hs00175721_m1; P2X4R, Hs00602442_m1; NLRP3, Hs00918082_m1; Caspase-1, Hs00354832_m1; AIM2, Hs00915710_m1; IL-2, Hs00174114_m1; IL-6, Hs00174131_m1; TNFα, Hs00174128_m1; TGFβ, Hs00998133_m1; GAPDH, Hs02758991_g1. The relative target gene expression, normalized to housekeeping gene GAPDH, is given as 2^−ΔΔCt^, where Ct is the threshold cycle and referred to their expression in six liver samples collected in patients with liver damage characterized by alterations of the ductal plate with minimal/absent necro-inflammation.

### 4.5. Immunofluorescence

Liver FFPE sections, after deparaffination, rehydration, and antigen retrieval steps, were processed for a standard immunostaining protocol. Briefly, to block nonspecific antibody binding, sections were washed three times with PBS pH 7.4 (10 min each) and treated with 5% BSA/0.3% Triton X-100 in PBS for 1 h. Samples were left to react overnight at 4 °C with different dilutions of specific primary antibodies, listed in Appendix A. After 1h incubation at room temperature with a secondary antibody conjugated with 568- (goat anti-rabbit) or 488-alexaFluo (goat-anti mouse), the slides were covered with an antifade mounting medium (ProLong Gold, Life Technologies, Monza, Italy), were acquired using a laser scanning confocal microscope (Leica TCS SP8, Milan, Italy). In detail, 20–30 images were acquired every 0.25/0.35 µm along the *z*-axis, sequentially for each channel, and the presence of double-stained cells was verified in every single focal plane; percentages of AIM2+ or NRLP3+ cells in CD68+ cells or P2X7R+ cells was also counted.

### 4.6. Interleukin-2 Levels

Circulating levels of IL-2 were measured in duplicate on plasma aliquots by a commercial ELISA (BMS 221-HS, Thermo Fisher).

### 4.7. Statistical Analysis

Continuous variables are reported as median and interquartile ranges, and categorical variables are reported as count and percentage. Groups were compared by Kruskal–Wallis test or Fisher exact test, respectively. Multivariable linear regression analysis was used to test the effect of liver disease on gene expression while controlling for potential confounders, including age, sex, and BMI. Multiple post-hoc pairwise comparisons were performed by the Steel–Dwass test. Linear associations between variables were tested using Spearman correlation analysis. Statistical tests were performed using JMP Pro 14 software (SAS, Cary, NC, USA). A *p*-value ≤ 0.05 was considered statistically significant.

## Figures and Tables

**Figure 1 ijms-23-07447-f001:**
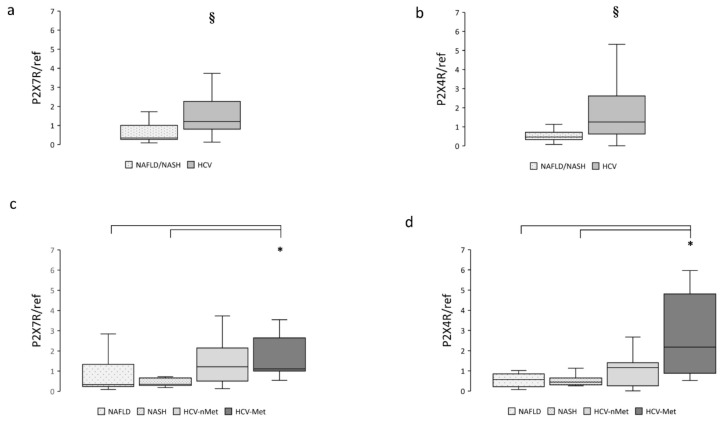
Hepatic P2X7R and P2X4R gene expression in the main study groups (**a**,**b**) and the four subgroups (**c**,**d**). Data are normalized to GAPDH expression. § *p* < 0.001, ***** *p* < 0.01.

**Figure 2 ijms-23-07447-f002:**
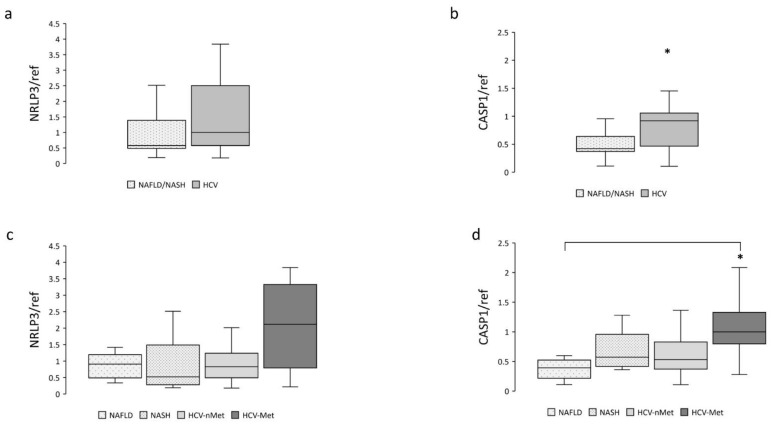
Hepatic NRLP3 and CASP1 gene expression in the main study groups (**a**,**b**) and the four subgroups (**c**,**d**). Data are normalized to GAPDH expression. ***** *p* < 0.05.

**Figure 3 ijms-23-07447-f003:**
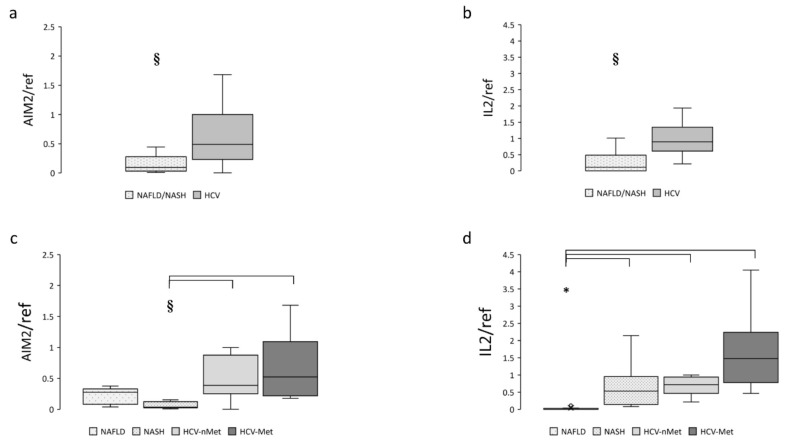
Hepatic AIM2 and IL-2 gene expression in the main study groups (**a**,**b**) and the four study subgroups (**c**,**d**). Data are normalized to GAPDH expression. § *p* < 0.005, ***** *p* < 0.0001.

**Figure 4 ijms-23-07447-f004:**
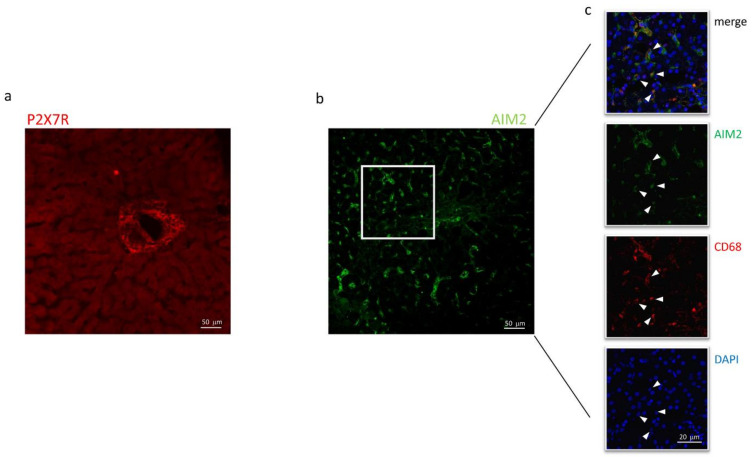
Confocal microscopy of liver biopsies shows P2X7R-immunostaining (red) surrounding the centrilobular vein (**a**) and AIM2-immunostaining (green), presumably near sinusoids (**b**). White arrows indicate double-stained cells. Confocal acquisitions of double staining experiments confirm AIM2 expression in Kupffer (CD68^+^) cells (**c**). White square in (**b**) points out the portion of the image analyzed in (**c**). The image is representative of experiments performed in 6 biopsies for each group; P2X7R and AIM2 localization was confirmed irrespective of the liver disease.

**Figure 5 ijms-23-07447-f005:**
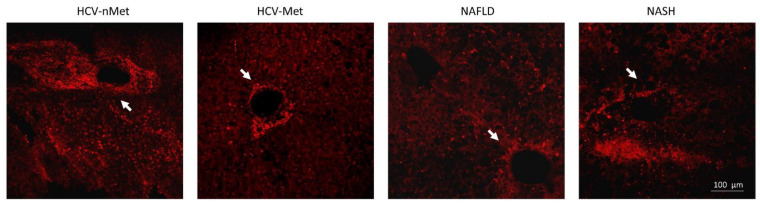
P2X7R immunoreactivity in the four groups of patients: in the presence of HCV infection, P2X7R surrounds vessels with a fibrillar aspect; in NAFLD/NASH groups, it appears more widespread in the tissue. White arrows indicate a centrilobular vein, where P2X7R immunoreactivity is evident. The image is representative of experiments performed in 6 biopsies for each group.

**Figure 6 ijms-23-07447-f006:**
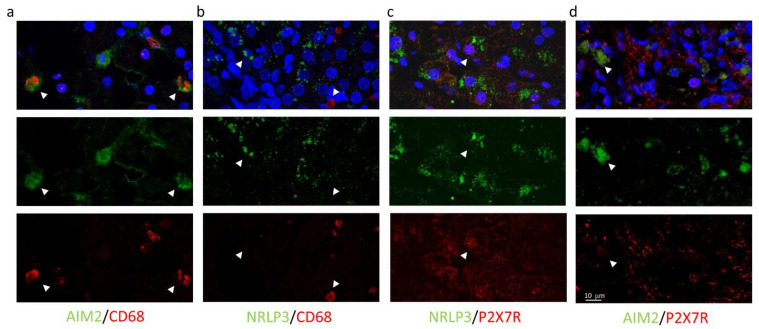
Confocal images of liver biopsies treated with the combination of primary antibodies confirm the expression of AIM2 (but not NRLP3) in CD68^+^ cells (**a**,**b**); conversely, P2X7R colocalizes with NRLP3 inflammasome (but not with AIM2) (**c**,**d**). In the first line, multicolor images: merge of primary antibodies and DAPI nuclear staining. In the second and third line: green and red indicate primary antibodies. Magnification 40×, Zoom2.

**Figure 7 ijms-23-07447-f007:**
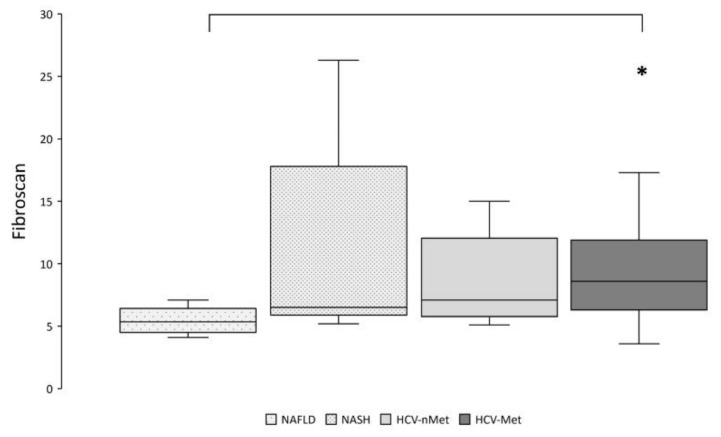
Hepatic fibrosis was assessed by transient elastography (Fibroscan^®^). * *p* < 0.01 by pairwise analysis.

**Figure 8 ijms-23-07447-f008:**
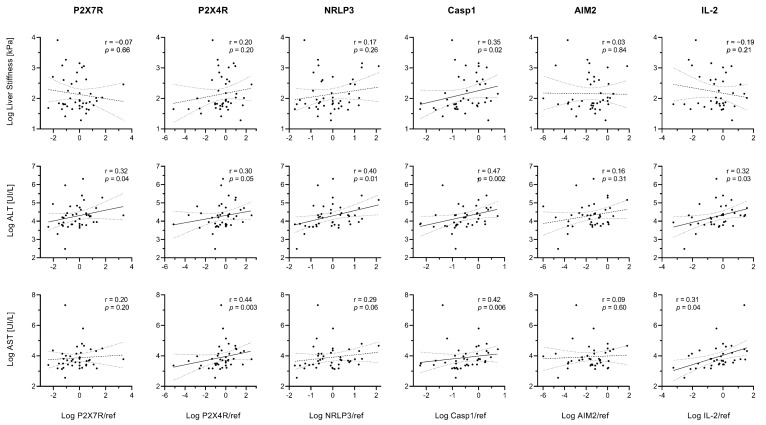
Relationships between gene expression and markers of disease severity. Spearman’s rank correlation coefficients (r) and *p* values (*p*) are reported. Data were log-transformed for graphical purposes. Best-fit lines of log-transformed data are shown as continuous lines (*p* < 0.05) or dashed lines (*p* ≥ 0.05) along with their 95% confidence bands (dotted lines).

**Table 1 ijms-23-07447-t001:** Anthropometric and biochemical characteristics of study participants.

	NAFLD/NASH			HCV				
	All (*n* = 21)	NAFLD (*n* = 10)	NASH (*n* = 11)	All (*n* = 25)	HCV-nMet (*n* = 12)	HCV-Met (*n* = 13)	*p* *	*p* **
Age	46.1 [38.5–53.2]	38.5 [34.9–47.0] ^a^	51.3 [44.4–61.7]	43.9 [36.8–47.9]	43.3 [31.2–45.6]	43.9 [41.1–52.3] ^a^	0.26	0.006
Men (*n*, %)	16, 76.2	9, 90.0	7, 63.6	21, 84.0	9, 75.0	12, 92.3	0.71	0.27
BMI (kg/m2)	24.6 [24.5–30.3]	24.8 [24.5–27.3]	28.7 [24.5–31.8]	24.5 [24.5–27.1]	24.5 [24.5–27.7]	24.7 [24.5–26.9]	0.58	0.30
Glucose (mg/dL)	85 [75–115]	83 [75–111]	92 [79–144]	84 [70–88]	81 [71–86]	87 [66–98]	0.14	0.41
Insulin (µU/mL)	14 [9–22]	13 [7–18]	16 [11–35]	7 [5–11]	9 [3–16]	7 [6–9]	0.03	0.18
Liver steatosis (kPa)	6 [5–12]	5 [5–6]	10 [6–23]	8 [6–12]	7 [6–12]	9 [6–15] ^b^	0.14	0.02
AST (UI/L)	35 [26–49]	30 [25–80]	40 [31–48]	45 [31–82]	41 [30–74]	56 [36–96]	0.08	0.21
ALT (UI/L)	52.5 [41–77]	50 [41–92]	67 [38–76]	74 [46–126]	73 [45–136]	74 [48–123]	0.08	0.37
GGT (U/L)	92 [36–206]	133 [47–235]	57 [34–129]	49 [25–89]	33 [22–151]	60 [45–89]	0.09	0.11
Total bilirubin (mg/dL)	0.68 [0.57–0.85]	0.72 [0.47–1.05]	0.68 [0.59–0.74]	0.95 [0.54–1.29]	0.65 [0.49–0.99]	1.04 [0.68–1.42]	0.26	0.17
Direct bilirubin (mg/dL)	0.22 [0.18–0.27]	0.18 [0.13–0.30]	0.25 [0.19–0.27]	0.22 [0.18–0.37]	0.21 [0.14–0.22]	0.34 [0.20–0.40]	0.44	0.11
Platelets (1000/μL)	219 [168–267]	266 [193–300]	184 [160–236]	207 [170–238]	199 [166–256]	208 [190–226]	0.55	0.21
Triglycerides (mg/dL)	108 [90–220]	179 [90–220]	102 [86–199]	84 [69–128]	87 [60–136]	81 [74–157]	0.16	0.52
HDL-cholesterol (mg/dL)	45 [40–50]	49 [43–57]	41 [40–45]	41 [29–46]	39 [37–40]	43 [26–47]	0.24	0.27
LDL-cholesterol (mg/dL)	124 [99–178]	160 [146–184]	100 [97–124]	97 [66–135]	119 [117–121]	82 [74–124]	0.19	0.21
Albumin (mg/dL)	4.7 [4.2–4.8]	4.8 [4.3–4.9]	4.6 [4.2–4.8]	4.6 [4.4–4.7]	4.4 [4.3–4.8]	4.6 [4.5–4.7]	0.64	0.86
INR	1.06 [1.00–1.12]	1.01 [0.97–1.07]	1.11 [1.05–1.13]	1.03 [1.01–1.08]	1.03 [0.98–1.06]	1.03 [1.01–1.12]	0.62	0.15
HCV RNA (UI/mL × 1000)	–	–	–	586 [171–6825]	253 [115–1623]	936 [261–1023]	-	0.07

Data are reported as mean ± SD or median [range]. Abbreviations: BMI, Body Mass Index; AST, Aspartate Transaminase; ALT, Alanine Transaminase; GGT, gamma-glutamyl transferase; INR, International Normalized Ratio. * *p*-value for the difference between the two main groups (NAFLD/NASH vs. HCV). ** *p*-value for the difference between the four subgroups (NAFLD, NASH, HCV-nMet, HCV-Met). ^a^ *p* < 0.05 for the difference with the NASH subgroup. ^b^ *p* < 0.05 for the difference with the NAFLD subgroup.

## Data Availability

Datasets used and analyzed during the current study will be available upon acceptance from the corresponding author on reasonable request.

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
