# Peer review of "The P2X7R-NLRP3 and AIM2 Inflammasome Platforms Mark the Complexity/Severity of Viral or Metabolic Liver Damage"

_ijms, 2022, doi:10.3390/ijms23137447_

Round 1

Reviewer 1 Report

Although the authors revised the manuscript to strengthen their conclusion, some noticeable weaknesses still existed.

1) In Fig. 4, confocal images with a thin optical thickness (7 μm) do not necessarily imply that both are in the same focal plane. Axial resolution should be considered. In addition, there were more than 4 cells showing the yellow fluorescence (potentially co-localization) while only 4 arrows were presented in the figure.

2) In Fig. 5, strong red fluorescence was also observed in the non-centrilobular area. Just manipulating the brightness and/or contrast would not strengthen the conclusion.

3) Although the authors put some efforts not to overstate the conclusion, the abstract and result were almost identical to the original manuscript.

Reviewer 2 Report

The authors resubmitted the manuscript, in which they revised the previous one with addressing the reviewer's comments appropriately.
